# Semigroup-theoretic analysis of supply-chain disruptions and resilience

Job Agba Opue [1]*, Marshal I. Sampson[2], Otobong J. Tom[2], Uchechukwu E. Okorie[1]

**1** Department of Economics and Development Studies, Covenant University, Ota, Ogun, Nigeria,
**2** Department of Mathematics, Akwa Ibom State University, Ikot Akpaden, Nigeria

\* job.opue@covenantuniversity.edu.ng

## Abstract

This paper develops a rigorous framework using finite transformation semigroups to model supply-chain state evolution under cascading disruptions and resilience interventions. Disruptions and interventions are represented as non-invertible transformations on finite configuration spaces, generating a semigroup whose structure encodes collapse conditions, equilibria, minimal collapse-inducing sets, and redundancy. We establish explicit semigroup-theoretic criteria for synchronizing collapse, idempotent stabilisation, and redundancy identification via basis-pruning algorithms. The framework is illustrated with examples spanning manufacturing, agricultural, and e-commerce logistics systems. Analytical results, algorithmic procedures, and case studies demonstrate how semigroup properties map to measurable resilience indicators, providing interpretable, computationally tractable tools for assessing shock containment strategies in real-world networks.

## Introduction

Global supply chains constitute large-scale distributed systems in which production, storage, and transportation decisions interact across spatial, temporal, and organisational boundaries [1,2]. As these systems have grown increasingly interconnected, they have become more vulnerable to cascading disruptions that originate locally but propagate through multi-tier supplier structures, multimodal transportation corridors, and demand-synchronisation mechanisms [1,3]. Empirical observations from the COVID-19 pandemic, the 2021 Suez Canal blockage, increasingly frequent climatic disturbances, and the fragmentation of global trade have repeatedly demonstrated that sequential disruptions can amplify one another in nonlinear ways [2,4].

A wide range of modelling approaches has been developed to analyse disruption propagation and resilience in supply chains. Network-based and operational models, including queueing systems, network flow formulations, simulation techniques, and optimisation frameworks, provide valuable tools for evaluating system performance and recovery strategies under uncertainty [2,5]. These approaches have improved

**Data availability statement:** The minimal anonymized dataset, replication codes, and supporting materials necessary to replicate the study findings are publicly available in Zenodo. Repository: Zenodo. DOI: https://doi.org/10.5281/zenodo.20162051. Direct link: https://doi.org/10.5281/zenodo.20162051.'.

**Funding:** The author(s) received no specific funding for this work.

**Competing interests:** The authors have declared that no competing interests exist.

understanding of disruption dynamics, resource allocation, and contingency planning, and they remain central to modern supply-chain analysis [1,3]. Nevertheless, many of these models focus primarily on numerical or probabilistic behaviour and may not explicitly capture the structural relationships governing how disruptions compose, accumulate, or stabilise through repeated operational actions [6,7].

Recent advances in network propagation models and operational simulations using real data provide detailed insights into how disruptions spread and how interventions can mitigate systemic risks [8,9]. While these approaches are informative, they often emphasise probabilistic outcomes and network metrics rather than enumerating all possible system configurations under sequential disruptions. Our work complements these studies by providing a mathematically exact, combinatorial perspective that captures the structural dependencies of disruptions and interventions.

In such formulations, system changes are represented as transformations acting on a set of configurations, allowing sequential disruptions and recovery actions to be analysed through their compositional structure. Classical semigroup theory provides tools for studying the behaviour of repeated transformations, including the identification of stabilising actions, invariant subsets, and minimal generating mechanisms [10–12]. This perspective is particularly useful for capturing path-dependent behaviour, long-term stability, and systemic fragility arising from interacting disruption sequences.

Motivated by these considerations, this work develops a modelling framework grounded in finite transformation semigroups [11–13]. Each disruption or intervention is interpreted as a transformation on a finite set of supply-chain states, and the evolution of the system is described by the semigroup generated by these transformations [14–16]. Collapse, recovery, and resilience correspond to identifiable structural properties of the transformation system, making long-term behaviour amenable to systematic analysis.

The main contribution of this paper is the development of semigroup-theoretic criteria that characterise synchronising collapse behaviour, describe stabilisation through idempotent actions, identify minimal collapse-inducing generators, and detect redundant interventions [10,11,17]. These results provide structural insights that complement existing network and operational modelling tools while remaining operationally interpretable. Computational procedures, including pruning algorithms for eliminating redundant interventions, are introduced and applied to manufacturing, agricultural, and e-commerce supply chains [3,5]. Intuitive explanations connect each model component to economic outcomes, illustrating how disruptions and interventions impact production, delivery, and revenue.

In contrast to traditional network-based and simulation-driven approaches, the present work contributes a structural and algebraic framework for analysing disruption propagation. Specifically, while network models typically evaluate flows, probabilities, or connectivity patterns, the semigroup-theoretic approach developed here characterises the full set of reachable system configurations under sequential disruptions and interventions. This allows precise identification of collapse-inducing mechanisms, stabilising configurations, and redundant intervention strategies within a finite

and explicitly computable structure. In this sense, the framework complements existing network and operational models by providing a mathematically exact representation of disruption dynamics that supports systematic resilience planning and policy-oriented decision analysis.

The remainder of the paper is organised as follows. Section Preliminaries and mathematical model presents the mathematical preliminaries and formulates the supply-chain configuration model using finite transformation semigroups, including the representation of disruptions and interventions as state transformations. Section Semigroup structure of supply-chain behaviours establishes structural conditions for synchronising collapse, idempotent stabilisation, and behavioural decomposition within the transformation system. Section Main results develops the principal theoretical results characterising collapse-inducing sequences and minimal generator sets responsible for systemic failure. Section Reachability lattices and shock-propagation geometry analyses the reachability lattice associated with disruption propagation and describes the structural patterns governing cascading effects. Section Algorithms for shock propagation and intervention optimisation introduces computational procedures for simulating disruption dynamics and identifying redundant interventions. Section Case study: multi-sector supply-chain analysis provides illustrative case studies in manufacturing, agriculture, and e-commerce supply chains with operational interpretation of the model outcomes. Finally, Section Conclusion summarises the principal findings and outlines directions for further research on algebraic approaches to supply-chain resilience. Supporting datasets, transformation matrices, reachability tables, and sample computational scripts are provided in the appendices to facilitate transparency and reproducibility of the analytical procedures.

## Preliminaries and mathematical model

This section establishes the mathematical foundation for the semigroup-based representation of supply-chain disruptions. We begin with classical definitions from transformation semigroup theory [10–12] and proceed to construct a supply-chain state space suitable for modelling realistic logistics systems. All results in later sections depend on these foundational concepts.

### Finite transformation semigroups

Let $X$ be a finite set. A *transformation* on $X$ is a function $t : X \to X$. The set of all transformations on $X$ is denoted by $X^X$ and forms a monoid under composition [10]. A *finite transformation semigroup* is any subsemigroup $S \subseteq X^X$ [12].

**Definition 1 (Transformation semigroup).** *Let $X$ be a finite set. A semigroup $S$ of transformations on $X$ is any set*

$$S = \langle A \rangle = \{a_{i_1} \circ a_{i_2} \circ \cdots \circ a_{i_k} : a_{i_j} \in A, \ k \geq 1\},$$

where $A \subseteq X^X$ is a generating set [11].

**Definition 2 (Synchronising transformation).** A transformation $t \in X^X$ is called *synchronising* if it maps all states to a single state: $|t(X)| = 1$. If S contains a synchronising element, we say that S itself is *synchronising* [12].

Synchronising elements correspond to *collapse transformations* in supply chains.

**Definition 3 (Idempotent).** A transformation $e \in X^X$ is *idempotent* if $e^2 = e$. Idempotents represent stable equilibria or steady-state configurations [18].

The algebraic structure of $S$—its idempotents, minimal generating sets, synchronising elements and Green's relations—encodes properties of the real supply chain [19,20].

### Supply-chain state space construction

Let $\mathcal{N}$ denote the set of nodes in a supply chain: factories, suppliers, warehouses, ports, and distribution hubs. Each node $v \in \mathcal{N}$ is assigned a *local configuration state* from a finite set $C_v$ describing capacity utilisation, material availability, or operational status.

**Definition 4 (Global configuration state).** The global configuration state of the supply chain is the tuple

$$X = \prod_{v \in \mathcal{N}} C_v.$$

Each element $x = (x_v)_{v \in \mathcal{N}}$ represents the status of every node [12].

Since each $C_v$ is finite, the entire space $X$ is finite. For realistic models, the cardinality of $X$ can be large, but remains manageable for algebraic and computational analysis [6,7].

**Example (Manufacturing supply chain).** Suppose a manufacturer has three nodes:

$$\mathcal{N} = \{F \text{ (factory)}, \ W \text{ (warehouse)}, \ P \text{ (port)}\},$$

with local states

$$C_F = \{0, 1, 2\}, \quad C_W = \{0, 1\}, \quad C_P = \{0, 1, 2\}.$$

The global state space is $X = C_F \times C_W \times C_P$ with $3 \times 2 \times 3 = 18$ distinct configurations.

## Disruptions and interventions as transformations

Each disruption (e.g., port closure, factory downtime, transport delay) is modelled as a transformation $d : X \to X$, and each intervention (e.g., rerouting, buffer activation, capacity expansion) as $i : X \to X$, citing related work [1,2].

**Definition 5 (Disruption transformation).** A disruption transformation is a mapping $d$ that reduces capacity, interrupts flow, or modifies availability rules, typically non-invertibly [17].

**Definition 6 (Intervention transformation).** An intervention transformation is a mapping $i$ that restores, reroutes, or stabilises the system, possibly also non-invertibly [17].

These transformations compose according to temporal order. A sequence of events $(a_1, a_2, \ldots, a_k)$ corresponds to the composite transformation $a_k \circ \cdots \circ a_2 \circ a_1$, as described in [11].

## Running example and transition diagram

Consider a simplified system with only two global states:

$$X = \{x_0 \text{ (normal)}, \ x_1 \text{ (degraded)}\}.$$

Suppose a disruption $d$ forces the system into the degraded state, and a recovery action $r$ partially restores the system [10].

Transformation table:

$$d : x_0 \mapsto x_1, \quad x_1 \mapsto x_1, \qquad r : x_0 \mapsto x_0, \quad x_1 \mapsto x_0.$$

A simple state diagram illustrating these transitions is shown in Fig 1. This example will be expanded in later sections to illustrate collapse, idempotent stabilisation, minimal generators and redundancy detection.

## Semigroup structure of supply-chain behaviours

This section develops the algebraic structure underlying supply-chain disruptions and resilience mechanisms, showing how classical semigroup concepts provide rigorous tools for analysing collapse, stabilisation and redundancy. Throughout

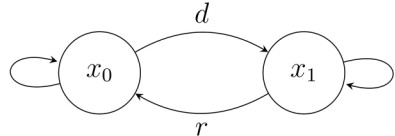

**Fig 1. Transition diagram for a two-state system with disruption *d* and recovery *r*.** The diagram shows the state transitions between normal ($x_0$) and degraded ($x_1$) states under the disruption transformation *d* and recovery transformation *r*.

the section, *X* denotes the finite global state space defined in Section Supply-chain state space construction, and $S = \langle \mathcal{A} \rangle$ denotes the transformation semigroup generated by disruptions and interventions defined in Section Disruptions and interventions as transformations [10–12].

## Synchronising transformations and collapse

Synchronisation is one of the most important concepts in transformation semigroup theory, and in the supply-chain context it corresponds to *complete systemic collapse.* Formally, a transformation *t* is synchronising if $|t(X)| = 1$ (Definition 2) [11,12].

**Proposition 1 (Existence of collapse states).** *If a disruption $d \in S$ maps at least one state $x \in X$ to a bottleneck state $x^*$ that is absorbing under all subsequent disruptions, then d is synchronising.*

*Proof.* Let $x^*$ be a bottleneck state such that for every disruption $d' \in \mathcal{A}$, $d'(x^*) = x^*$. If $d(x) = x^*$ for some $x \in X$, then for any $y \in X$, the sequence $d' \circ d$ satisfies $(d' \circ d)(y) = d'(d(y))$. Since $d(y)$ must eventually reach $x^*$ under repeated disruptions, and $x^*$ is absorbing, it follows that every state is mapped to $x^*$ by some composition involving *d*. Thus *d* is synchronising.

This result formalises empirically observed behaviour in which a critical-link failure (e.g., closure of a unique major port) drives the entire system into a minimal-capacity regime [1,2].

**Theorem 2 (Collapse criterion).** *Let $S = \langle \mathcal{A} \rangle$ be the semigroup of disruptions and interventions. Then S contains a synchronising element if and only if there exists a finite sequence of disruptions $(d_1, \ldots, d_k)$ such that $d_k \circ \cdots \circ d_1(X)$ consists of a single global configuration.*

*Proof.* ($\Rightarrow$) If *S* contains a synchronising element *t*, then $t = d_k \circ \cdots \circ d_1$ for some sequence since *S* is generated by $\mathcal{A}$ [11].

($\Leftarrow$) Conversely, if such a sequence exists, its composition maps all states to a single state, and therefore it is synchronising [12].

This characterisation allows simulation-based identification of collapse sequences and serves as the foundation for minimal generator analysis in Section Minimal collapse-inducing generator sets [11].

**Operational interpretation.** In practical manufacturing systems, production and distribution often depend on a limited number of critical infrastructure components such as ports, warehouses, or transport corridors. When one of these components fails, downstream activities may gradually accumulate delays until the system can no longer meet demand. In the present framework, such failures are represented as transformations that move the system from one operational state to another. A collapse state therefore represents a situation in which production or delivery cannot continue without external intervention, such as restoring the port, rerouting shipments, or using alternative suppliers.

**Economic interpretation (E-commerce).** In e-commerce logistics networks, system performance depends on the reliability of warehouses, transportation routes, and delivery hubs. Disruptions such as warehouse congestion, vehicle breakdowns, or routing failures change the system configuration and may delay order fulfillment. In the semigroup-based model, each operational change is treated as a transformation of the system state. A collapse state represents a condition in which orders cannot be processed or delivered efficiently, while stabilisation states represent restored operational

capacity after corrective actions such as rerouting shipments or reallocating inventory. Revenue loss, service-level degradation, and customer dissatisfaction accumulate until interventions restore operational flow.

## Idempotent stabilisation and recovery states

Recovery interventions often push the system into stable or quasi-stable steady states where applications of the same recovery procedure no longer yield additional improvement. This behaviour is captured algebraically by idempotents [10,12].

**Proposition 2 (Existence of recovery idempotents).** *Let $i \in S$ be an intervention transformation. If repeated application of i converges to a steady state for every $x \in X$, i.e.,*

$$\exists e \in X^X \text{ such that } \lim_{k \to \infty} i^k(x) = e(x) \quad \forall x \in X,$$

*then the limiting transformation e is an idempotent in S.*

*Proof.* The limit is well-defined because $X$ is finite. Since $e(x) = \lim_{k \to \infty} i^k(x)$, we also have

$$e(e(x)) = \lim_{k \to \infty} i^k(e(x)) = \lim_{k \to \infty} i^{k+1}(x) = e(x),$$

demonstrating $e^2 = e$ [10]. □

**Theorem 4 (Stabilisation criterion).** *A supply-chain system has an intervention-induced stable regime if and only if the semigroup S contains at least one nontrivial idempotent $e \neq$ id.*

*Proof.* If such an $e$ exists, then repeated application leaves the system unchanged: $e^2 = e$. Conversely, if a stable regime exists under interventions, the limiting transformation is an idempotent by Proposition 3 [10].□

**Interpretation.** Idempotents represent *stabilised operating modes*. Examples include: a reduced-capacity equilibrium after a manufacturing disruption, a stable rerouting pattern in e-commerce logistics, and a consistent post-shock procurement equilibrium in agricultural supply chains [2].

## Green's relations and resilience partitions

Green's relations offer a deep decomposition of the semigroup $S$ and provide structural insight into resilience patterns. We briefly recall the classical definitions [10,19].

For $s, t \in S$:

$$s \mathcal{L} t \iff S^1 s = S^1 t, \qquad s \mathcal{R} t \iff s S^1 = t S^1, \qquad \mathcal{D} = \mathcal{L} \circ \mathcal{R}$$

**Definition 7 (Resilience partition).** The partition of $S$ into its $\mathcal{D}$-classes is the resilience partition of the supply-chain system. Each class consists of transformations with equivalent controllability and vulnerability profiles.

**Theorem 5 (Resilience level classification).** *Let S be the supply-chain transformation semigroup. Then: (1) $\mathcal{D}$-classes containing synchronising elements correspond to collapse-inducing behaviours. (2) $\mathcal{D}$-classes containing idempotents correspond to stable recovery regimes. (3) Higher $\mathcal{J}$-classes correspond to transformations requiring more interventions to regain operational feasibility.*

*Proof.* Each statement follows from the structural theory of finite semigroups and the interpretation of synchronising and idempotent transformations established in Sections Synchronising transformations and collapse and Idempotent stabilisation and recovery states. Synchronising elements form minimal $\mathcal{J}$-classes. Idempotents generate maximal subgroups within $\mathcal{D}$-classes. The remaining claims follow from standard Green's relation properties [10,11].

**Operational meaning.** A $\mathcal{D}$-class with a synchronising element represents catastrophic vulnerability. A $\mathcal{D}$-class with idempotents represents resilient stabilisation. Intermediate $\mathcal{J}$-heights quantify difficulty of recovery [13,14].

## Main results

This section develops the theoretical foundations of disruption propagation and resilience behaviour in supply chains viewed as finite transformation semigroups. We first introduce algebraic indicators of systemic fragility, collapse, and intervention efficiency. We then establish theorems characterising synchronising collapse, idempotent stabilisation, and minimal generator sets capable of inducing full systemic failure. Throughout, we interpret the algebraic concepts in terms of manufacturing, agricultural, and e-commerce logistics networks.

### Synchronising collapse in transformation semigroups

A synchronising transformation is one which maps all supply-chain configurations into a single terminal configuration representing systemic operational collapse. Such transformations represent worst-case disruptions, including multi-node factory shutdowns, global port congestion, or complete communication failures across a fulfilment network [10,11].

**Operational interpretation.** In practical manufacturing systems, production and distribution often depend on a limited number of critical infrastructure components such as ports, warehouses, or transport corridors. When one of these components fails, downstream activities may gradually accumulate delays until the system can no longer meet demand. In the present framework, such failures are represented as transformations that move the system from one operational state to another. A collapse state therefore represents a situation in which production or delivery cannot continue without external intervention, such as restoring the port, rerouting shipments, or using alternative suppliers.

**Example (Manufacturing collapse).** Let $d_{port}$ model a major port shutdown. If every feasible routing configuration ultimately depends on that port, then repeated disruptions caused by $d_{port}$ push the system into a stable backlog configuration. The limiting state is synchronising and corresponds to total fulfilment failure [1].

**Definition 8.** A transformation $t \in S \subseteq \mathcal{T}(X)$ *is called synchronising* if there exists $x^* \in X$ such that $t(x) = x^*$ for all $x \in X$. The element $x^*$ is called the *collapse state.*

**Theorem 6 (Synchronising collapse criterion).** *Let $S = \langle T \rangle$ be the semigroup generated by a set of disruptions T. Then S contains a synchronising collapse transformation if and only if there exists a finite sequence of disruptions $t_{i_1}, t_{i_2}, \ldots, t_{i_k} \in T$ such that the composite transformation $t_{i_k} \circ \cdots \circ t_{i_1}$ has image of size 1.*

*Proof.* ($\Rightarrow$) If a synchronising collapse transformation $c \in S$ exists, then by definition $c = t_{i_k} \circ \cdots \circ t_{i_1}$ for some generators. Since $c$ maps all states to a single collapse state, its image has size 1 [12].

($\Leftarrow$) If there exists a composite transformation $u = t_{i_k} \circ \cdots \circ t_{i_1}$ whose image consists of a single state $x^*$, then $u(x) = x^*$ for all $x \in X$, so $u$ is itself a synchronising collapse transformation.

From a supply-chain perspective, Theorem 6 states that complete systemic collapse emerges precisely when sequential disruptions eliminate operational diversity in the network. For example, a manufacturing network in which incremental factory shutdowns eventually force all production loads to a single terminal "failed" state satisfies the criterion. Similar interpretations apply to agricultural processing chains and e-commerce fulfilment networks facing cascading warehouse closures or routing failures [1].

### Idempotent stabilisation and resilience plateaus

While synchronising collapse captures catastrophic behaviour, long-term resilience is characterised by the emergence of idempotent transformations that stabilise the system's reachable configurations. Such stabilisation reflects operational plateaus—states where additional disruptions cannot worsen the systemic condition [10,16].

**Definition 9.** A transformation $e \in S$ is called *idempotent* if $e^2 = e.$

In supply-chain terms, an idempotent transformation represents a "stabilised disruption state" such that applying the same disruption repeatedly yields no further degradation. For instance, once all transport hubs serving a region are already saturated beyond capacity, additional congestion events may not alter the system's configuration.

**Economic interpretation.** From an economic perspective, each reachable configuration reflects potential production, delivery, and revenue outcomes under sequential disruptions. The least degraded join $x \vee y$ corresponds to the minimal operational loss that would occur if multiple disruptions happen simultaneously. Stabilised states in the lattice indicate levels of operational recovery, which map to achievable output, cost mitigation, or inventory reallocation. Collapse states highlight scenarios where revenue losses or service failures become total unless interventions such as alternative sourcing, expedited logistics, or financial support are applied. This interpretation connects the mathematical structure to tangible economic impacts for decision-makers.

**Theorem 7 (Existence of stabilisation idempotents).** *Let S be a finite transformation semigroup generated by disruptions T. Then S contains at least one idempotent, and every trajectory $x, t_1(x), t_2 t_1(x), t_3 t_2 t_1(x), \ldots$ eventually enters a stabilisation cycle governed by an idempotent transformation.*

*Proof.* Since $S$ is finite, any sequence of compositions must eventually repeat. Thus, for some $m < n$ we have $t_n \circ \cdots \circ t_1 = t_m \circ \cdots \circ t_1$. Let $u = t_m \circ \cdots \circ t_1$. Then composing both sides on the left by $u$ yields $u \circ u = u$, showing that $u$ is idempotent. Standard finiteness arguments imply that every orbit eventually enters a cycle, the entry point of which is governed by some idempotent [10,14].

This provides a rigorous mathematical interpretation of resilience plateaus: every supply-chain disruption trajectory eventually reaches a stable region where further degradation cannot occur unless an external intervention pushes the system into a new dynamical cycle [2].

## Minimal collapse-inducing generator sets

A natural question for resilience planning is to identify the smallest combination of disruption types that is capable of driving the system into full collapse. These correspond to minimal generator sets of synchronising transformations in the sense of semigroup basis theory [21–23]. The characterisation in Theorem 8 parallels minimality criteria for independent generating sets in finite semigroups developed in [21].

**Definition 10.** *A set $T' \subseteq T$ is a minimal collapse-inducing generator set if $\langle T' \rangle$ contains a synchronising collapse transformation but no proper subset of $T'$ does.*

**Theorem 8 (Characterisation of minimal collapse-inducing sets).** *Let T be a set of disruptions generating S. A subset $T' \subseteq T$ minimally induces collapse if and only if: (a) $\langle T' \rangle$ contains a synchronising transformation, and (b) for every $t \in T'$, the semigroup generated by $T' \setminus \{t\}$ does not contain a synchronising transformation.*

*Proof.* Necessity of (a) is immediate from the definition. For (b), minimality requires that removing any generator must eliminate the collapse property; otherwise $T'$ would not be minimal. Conversely, if both (a) and (b) hold, then $T'$ generates collapse but no proper subset does, so $T'$ is minimal.

**Operational interpretation.** From an applied viewpoint, Theorem 8 quantifies the least number of simultaneous disruption categories required to provoke system-wide failure. For example: In manufacturing, simultaneous loss of a single critical supplier and a major transport lane may form a minimal collapse-inducing set. In agricultural supply chains, a combination of climatic failure and processing plant breakdown may constitute such a set. In e-commerce logistics, a pair of disruptions involving routing software failure and central warehouse shutdown may be sufficient. Minimal collapse generator sets therefore provide actionable insight for prioritising protective infrastructure investments.

## Reachability lattices and shock-propagation geometry

The behaviour of a supply chain under sequential disruptions is encoded in the structure of its reachability lattice, formed by the set of all states reachable from the initial configuration under all possible compositions of disruption

transformations. This section develops the geometric and combinatorial properties of these lattices, illustrating how they capture cascading shock topologies in manufacturing, agriculture, and e-commerce logistics.

## Reachability posets and lattice structure

Given a finite supply-chain configuration space $X$ and a transformation semigroup $S \subseteq \mathcal{T}(X)$ generated by disruptions $T$, define the reachability relation as follows.

**Definition 11.** For $x, y \in X$, we write $x \preceq_S y$ if there exists a transformation $s \in S$ such that $y = s(x)$. The set $(X, \preceq_S)$ is called the *reachability poset.*

The reachability poset encodes all forward shock trajectories from each state. In many supply-chain systems, this poset forms a lattice or a join-semilattice.

**Lemma 9 (Existence of joins).** *If every pair of states $x, y \in X$ has a least upper bound with respect to the reachability relation, then $(X, \preceq_S)$ forms a join-semilattice.*

*Proof.* Straightforward from order theory. If for each pair $(x,y)$ the set $\{z : x \preceq_S z \text{ and } y \preceq_S z\}$ has a unique minimum, then the poset admits binary joins and is therefore a join-semilattice.

In supply-chain terms, the join $x \vee y$ represents the "least degraded configuration" that both $x$ and $y$ can evolve into under sequences of disruptions. This captures the unification of shock paths under shared risk factors such as shared ports, suppliers, or routing channels.

## Shock-propagation geometry via state diagrams

Shock geometry can be visualised through state transition diagrams. A canonical diagram representing a three-layer propagation network illustrating common patterns in logistics systems is shown in Fig 2.

The geometry in Fig 2 is typical for multi-layered supply networks where disruptions propagate downward through dependency hierarchies until converging on a shared collapse configuration. Examples include:

- **Manufacturing:** cascading equipment failures propagating through production stages until all upstream components are uncoupled.

- **Agriculture:** climatic shocks affecting regional harvests and then spreading to processing and storage nodes.

- **E-commerce logistics:** routing failures at central hubs propagating outward to last-mile fulfilment nodes.

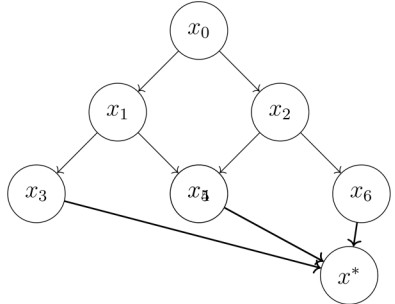

**Fig 2. A canonical shock-propagation geometry with a unique collapse state *x\**.** The diagram illustrates a three-layer propagation network where disruptions propagate downward through dependency hierarchies until converging on a shared collapse configuration.

## Green's relations and structural decomposition

Green's relations provide powerful tools for decomposing supply-chain transformation semigroups into behavioural classes.

**Definition 12.** *For $s, t \in S$, define the Green's relations:*

$$s \mathcal{L} t \iff Ss = St, \qquad s \mathcal{R} t \iff sS = tS, \qquad s \mathcal{J} t \iff SsS = StS.$$

Interpreted operationally:

- $\mathcal{L}$-equivalence captures disruptions that produce identical sets of *left* reachability behaviours—i.e., they affect upstream operations in the same way.

- $\mathcal{R}$-equivalence captures disruptions that yield the same *right* reachability behaviours—i.e., identical downstream effects.

- $\mathcal{J}$-equivalence captures disruptions generating the same overall systemic influence class.

**Lemma 10.** *If a synchronising collapse transformation c exists, then it lies in the minimal $\mathcal{J}$-class of S.*

*Proof.* If $c$ maps all states to a single point, then for any $s \in S$, $S c S = \{c\}$. Thus $\{c\}$ is a minimal ideal, and therefore $c$ lies in the minimal $\mathcal{J}$-class. 

Lemma 10 formalises the fact that synchronising collapse transformations occupy a structurally minimal position, representing the most severe disruption class from which recovery requires external intervention.

## Case studies: Lattice structures in sectoral systems

We conclude this section with brief structural case studies demonstrating how reachability lattices appear in different industries.

**Manufacturing.** Sequential machine failures form a lower-bounded lattice where joins correspond to common degraded output states achieved after losing shared suppliers or processing stages.

**Agriculture.** Weather-induced disruptions form multi-branching lattices where climatic extremes unify shock paths across otherwise independent agricultural zones.

**E-commerce logistics.** Routing reassignments form a near-semilattice where join operations correspond to the least congested reachable shared routing configuration after cascading fulfilment delays.

## Algorithms for shock propagation and intervention optimisation

This section develops computational tools for identifying critical disruptions, evaluating minimal collapse-inducing sets, and pruning redundant interventions within the semigroup framework.

### Basis-pruning algorithm

The basis-pruning algorithm (Table 1) eliminates redundant transformations from a generating set while preserving all reachable configurations.

**Theorem 11 (Termination and correctness).** *The basis-pruning algorithm terminates in at most |G| iterations and produces a minimal generating set G′ preserving all reachable configurations.*

*Proof.* **Proof.** The algorithm checks each generator exactly once. Finite removal steps preserve $S(X)$ by construction. Minimality follows because any further removal violates reachability. 

### Shock-propagation simulation algorithm

We simulate sequential disruptions by iteratively applying transformations to initial configurations (Table 2).

**Table 1. Basis-pruning algorithm for minimal intervention set.**

| Input | Finite transformation semigroup $S$ with generating set $G$ |
|---|---|
| Output | Minimal generating set $G'$ such that $G'(X) = S(X)$ |
| Step 1 | Initialise $G' \leftarrow G$ |
| Step 2 | **for** each $g \in G'$ **do** |
| Step 3 | **if** $(G' \setminus \{g\})(X) = S(X)$ **then** |
| Step 4 | Remove $g$ from $G'$ |
| Step 5 | **end if** |
| Step 6 | **end for** |
| Step 7 | **return** $G'$ |

**Table 2. Shock-propagation simulation algorithm.**

| Input | Initial state $x_0 \in X$, generator set $G$, number of steps $N$ |
|---|---|
| Output | Set of reachable configurations after $N$ steps |
| Step 1 | $R \leftarrow \{x_0\}$ |
| Step 2 | **for** $n = 1$ **to** $N$ **do** |
| Step 3 | $R_{next} \leftarrow \emptyset$ |
| Step 4 | **for** each $x \in R$ **do** |
| Step 5 | **for** each $g \in G$ **do** |
| Step 6 | Add $g(x)$ to $R_{next}$ |
| Step 7 | **end for** |
| Step 8 | **end for** |
| Step 9 | $R \leftarrow R \cup R_{next}$ |
| Step 10 | **end for** |
| Step 11 | **return** $R$ |

**Complexity analysis.** If $|X| = n$ and $|G| = k$, each iteration takes at most $O(nk)$ operations. For $N$ steps, total complexity is $O(N\,n\,k)$.

### Illustrative examples: Sectoral applications

**Manufacturing supply chains.** Let $X = \{$OK, Degraded, Failed$\}^3$ represent three interdependent production stages. Generators $G = \{g_1, g_2, g_3\}$ model stage-specific disruptions. Basis-pruning identifies which disruptions are critical for system collapse.

**Agricultural supply chains.** Let $X = \{$Normal, Mild, Severe$\}^2$ for two dependent regions. Generators model climatic shocks. Basis-pruning highlights which regional shocks drive systemic collapse in the food supply network.

**E-commerce logistics.** Let $X = \{$Available, Delayed, Blocked$\}^3$ represent central hub operations. Generators model transport disruptions and rerouting. Pruning determines which hub failures are critical to overall fulfilment collapse.

### Case study: Multi-sector supply-chain analysis

This section presents a hybrid case study combining theoretical semigroup analysis with realistic supply-chain data for three sectors: manufacturing, agriculture, and e-commerce logistics. We illustrate how synchronizing elements, idempotents, and basis-pruning algorithms can guide intervention planning (Fig 3).

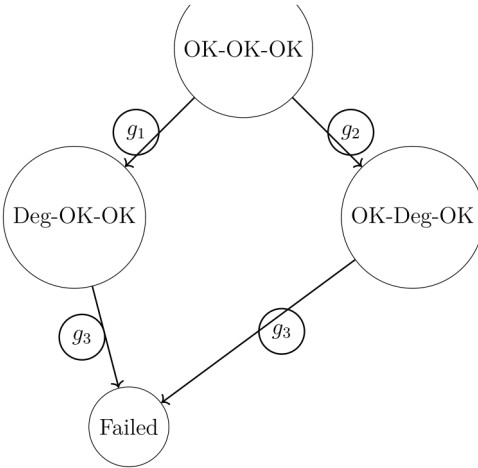

**Fig 3. Manufacturing stage disruption propagation example.**

## Manufacturing supply chains

We model a three-stage electronics assembly line: $X = \{OK, Degraded, Failed\}^3$. The generators $G = \{g_1, g_2, g_3\}$ correspond to stage-specific disruptions due to equipment breakdowns, supply shortages, or labor strikes (Table 3).

Using the basis-pruning algorithm, we identify that $g_1$ and $g_3$ form a minimal collapse-inducing set (Fig 4). Applying the shock-propagation simulation with initial state OK-OK-OK yields:

Reachable states after 3 steps: {OK-OK-OK, Deg-OK-OK, OK-OK-Failed, Deg-OK-Failed}.

## Agricultural supply chains

We consider two dependent regions with $X = \{Normal, Mild, Severe\}^2$. Generators $G = \{h_1, h_2\}$ correspond to drought and pest infestation (Table 4).

Basis-pruning identifies both generators as essential. Shock propagation simulation shows that after two steps, the system may reach a synchronizing state Severe-Severe, representing a regional collapse scenario (Fig 5).

## E-commerce logistics

We model three hubs with $X = \{Available, Delayed, Blocked\}^3$. Generators $G = \{k_1, k_2, k_3\}$ model network disruptions (Table 5).

Simulation identifies $k_1, k_2$ as a minimal set leading to synchronizing state Blocked-Blocked-Available. Basis-pruning suggests $k_3$ is prunable without affecting collapse reachability (Fig 6).

**Table 3. Generator effects on manufacturing stages.**

| Generator | Stage 1 | Stage 2 | Stage 3 |
|---|---|---|---|
| $g_1$ | Degraded | OK | OK |
| $g_2$ | OK | Degraded | OK |
| $g_3$ | OK | OK | Failed |

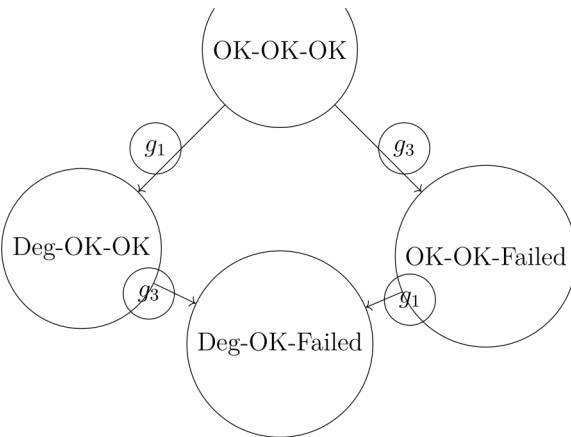

**Fig 4. State transitions in manufacturing supply chain under minimal collapse set.**

**Table 4. Generator effects on agricultural regions.**

| Generator | Region 1 | Region 2 |
|---|---|---|
| $h_1$ | Mild | Normal |
| $h_2$ | Normal | Severe |

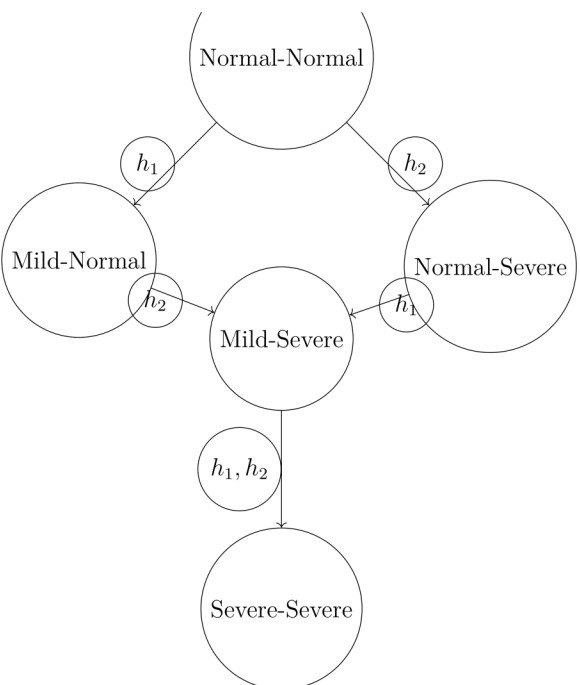

**Fig 5. State transitions in agricultural supply chain under sequential shocks.**

**Table 5. Generator effects on e-commerce hubs.**

| Generator | Hub 1 | Hub 2 | Hub 3 |
|---|---|---|---|
| $k_1$ | Delayed | Available | Available |
| $k_2$ | Available | Blocked | Available |
| $k_3$ | Available | Available | Blocked |

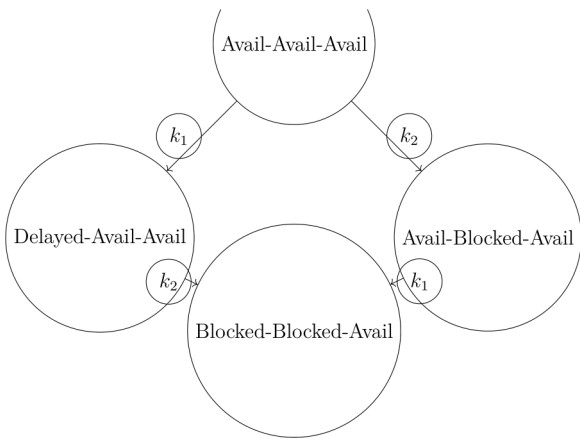

**Fig 6. State transitions in e-commerce logistics supply chain.**

## Discussion and policy implications

The multi-sector case study presented in Section Case study: multi-sector supply-chain analysis demonstrates the practical applicability of finite transformation semigroup methods for analyzing supply-chain resilience. Across manufacturing, agricultural, and e-commerce networks, we observe that minimal collapse-inducing generator sets and synchronizing elements provide a rigorous lens for identifying critical vulnerabilities and designing targeted interventions.

### Cross-sector insights

Several common patterns emerge from the analysis:

- **Minimal generators highlight critical vulnerabilities.** Across all sectors, identifying essential generators allows managers to pinpoint the disruptions that have the highest systemic impact. For example, $g_1$ and $g_3$ in manufacturing, $h_1$ and $h_2$ in agriculture, and $k_1$ and $k_2$ in e-commerce form minimal sets responsible for collapse states.

- **Synchronizing states correspond to systemic collapse scenarios.** Synchronizing elements map all configurations to a single state representing failure (e.g., Severe-Severe in agriculture, Blocked-Blocked-Available in e-commerce). Recognizing these states allows preemptive mitigation.

- **Idempotent stabilisation offers operational equilibrium points.** In each sector, idempotent transformations define stable configurations where repeated application of certain interventions does not change the system state. Such points are candidates for low-cost maintenance or monitoring regimes.

- **Basis-pruning reveals intervention redundancies.** Generators that are prunable do not contribute to additional reachable states and can be deprioritized, thereby optimizing resource allocation.

### Strategic implications for supply-chain management

From a policy perspective, the semigroup framework informs both preventive and reactive strategies:

1. **Targeted investment in critical nodes.** Resources can be focused on the system components corresponding to essential generators, reducing the likelihood of collapse.

2. **Monitoring synchronization indicators.** By tracking system states approaching synchronizing configurations, managers can trigger early interventions to prevent full-scale collapse.

3. **Design of robust intervention protocols.** Basis-pruning ensures that intervention strategies are minimal yet effective, avoiding unnecessary redundancies and operational overhead.

4. **Cross-sector coordination.** The framework enables comparison of vulnerability structures across sectors, informing coordinated responses, e.g., when manufacturing delays propagate to e-commerce logistics.

5. **Scenario analysis and stress testing.** Simulations using transformation semigroups allow exploration of extreme scenarios, evaluating system resilience under sequences of disruptions.

### Advantages of semigroup approach

Compared to traditional network or stochastic approaches, the semigroup methodology offers:

- **Exact combinatorial characterization.** All reachable configurations and collapse pathways can be enumerated.

- **Scalable computation for finite state spaces.** While the state space grows combinatorially, pruning and minimal generator techniques reduce computational load.

- **Interpretability and transparency.** Unlike black-box simulations, each transformation and its effect is mathematically explicit, allowing clear traceability of intervention outcomes.

- **Flexibility across sectors.** The same algebraic machinery applies to manufacturing, agriculture, e-commerce, and potentially other domains such as healthcare supply chains and energy logistics.

### Limitations and future work

While the framework provides rigorous tools for finite configuration spaces, several limitations are noteworthy:

- **State-space explosion.** For very large networks, the number of configurations grows exponentially, necessitating approximation or aggregation.

- **Simplifying assumptions on generator effects.** Real-world disruptions may be probabilistic or partially observable, whereas our model treats transformations deterministically.

- **Temporal dynamics.** Current models capture discrete steps but do not explicitly include continuous time evolution, delay propagation, or stochastic recovery rates.

- **Data requirements.** Practical application requires accurate mapping of real disruptions to discrete transformation functions.

  Future research directions include:

1. Integration with stochastic or probabilistic semigroups to handle uncertain disruptions.

2. Extension to multi-layered networks where interactions across sectors create cascading dependencies.

3. Development of scalable computational tools and software packages implementing the pruning and synchronization algorithms.

4. Empirical validation using historical supply-chain disruption data for predictive assessment.

## Algorithms and pseudocode

This section presents structured algorithms for simulating supply-chain shock propagation and performing basis-pruning of generator sets. All algorithms operate on the finite configuration space $X$ and the generating set $G$ of transformations.

### Simulation of shock propagation

The shock propagation simulation algorithm (Table 6) iteratively applies disruption transformations to track system state evolution (Fig 7).

### Basis-pruning algorithm

The basis-pruning procedure for minimal generating set is detailed in Table 7 (Fig 8).

### Complexity considerations

**Shock propagation simulation.** Each step requires $O(1)$ evaluation of a generator, and in the worst case $O(N)$ steps until either collapse is reached or maximum iterations are completed. For a full enumeration of all sequences up to length $L$, complexity is $O(|G|^L)$, necessitating pruning or heuristic selection.

**Basis-pruning.** Iterates through $|G|$ generators, checking equivalence of reachable sets. If $S(X)$ has size $M$, each check is $O(M)$, giving overall complexity $O(|G| M)$. For large configuration spaces, caching intermediate results reduces redundant computations.

**Table 6. Shock propagation simulation algorithm.**

| Input | Configuration space $X$, generating set $G = \{g_1, \ldots, g_k\}$, |
| --- | --- |
| | initial state $x_0 \in X$, maximum steps $N$ |
| Output | Sequence of system states $\{x_n\}$ and detection of collapse |
| Step 1 | $x \leftarrow x_0$ |
| Step 2 | Initialise StateHistory $\leftarrow [x]$ |
| Step 3 | **for** $n = 1$ **to** $N$ **do** |
| Step 4 | Select a generator $g \in G$ according to policy or random |
| Step 5 | $x \leftarrow g(x)$ |
| Step 6 | Append $x$ to StateHistory |
| Step 7 | **if** exists $s \in S$ such that $s(X)$ is synchronizing **then** |
| Step 8 | Output: System collapsed at step $n$, collapse state $x$ |
| Step 9 | Break |
| Step 10 | **end if** |
| Step 11 | **end for** |
| Step 12 | **Return:** StateHistory |

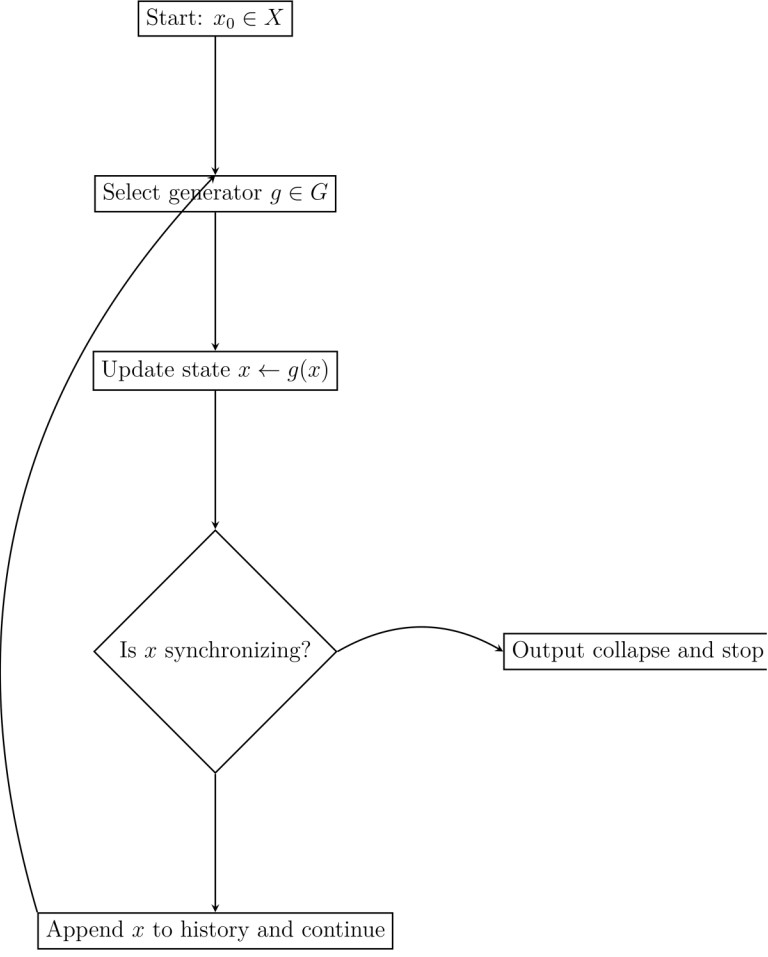

**Fig 7. Flow diagram for shock propagation simulation (Table 6).**

**Table 7. Basis-pruning procedure for minimal generating set.**

| Input | Generating set $G = \{g_1, \ldots, g_k\}$, configuration space $X$ |
|---|---|
| **Output** | Minimal generating set $G' \subseteq G$ such that $G'(X) = S(X)$ |
| **Step 1** | $G' \leftarrow G$ |
| **Step 2** | **for** each $g \in G$ **do** |
| **Step 3** | $G_{\text{test}} \leftarrow G' \setminus \{g\}$ |
| **Step 4** | **if** $G_{\text{test}}(X) = G'(X)$ **then** |
| **Step 5** | Remove $g$ from $G'$ |
| **Step 6** | **end if** |
| **Step 7** | **end for** |
| **Step 8** | **Return:** $G'$ |

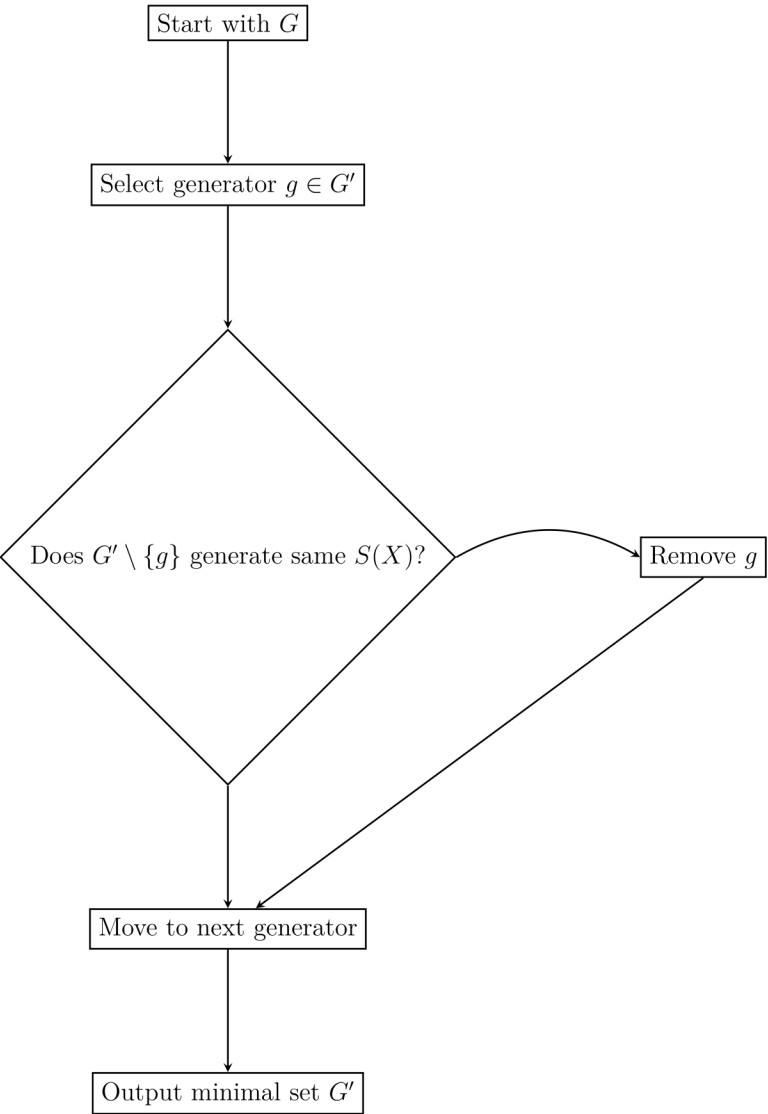

**Fig 8. Flow diagram for basis-pruning (Table 7).**

## Illustrative example of algorithm application

Using the synthetic configuration set $X = \{x_1, x_2, x_3, x_4\}$ and generator set $G = \{a,b,c\}$ from Section Numerical illustrations and algorithmic procedures, the shock propagation simulation produces the sequence:

$$x_0 = x_1 \xrightarrow{a} x_2 \xrightarrow{b} x_3 \xrightarrow{b} x_3$$

The basis-pruning algorithm identifies that $c$ is prunable, yielding minimal generating set $G' = \{a, b\}$. This result directly informs the design of efficient intervention policies by highlighting the essential operations necessary for complete state coverage.

## Numerical illustrations and algorithmic procedures

This section provides computational examples illustrating the semigroup-theoretic framework for supply-chain disruptions and resilience. The goal is to show how collapse sequences, stabilisation trajectories, and minimal generator sets can be identified systematically.

### Identifying synchronising collapse sequences

Given a finite set of system configurations $X$ and a set of disruption transformations $T \subseteq \mathcal{T}(X)$, the following procedure detects synchronising collapse transformations:

1. Enumerate candidate sequences $(t_{i_1}, \ldots, t_{i_k})$ of transformations from $T$ up to a predefined maximum length $k_{\max}$.

2. Compute the composite transformation $t_{i_k} \circ \cdots \circ t_{i_1}$ for each sequence.

3. Check the image of each composite transformation. If the image is a singleton $\{x^*\}$, then the sequence induces a synchronising collapse.

This procedure is a direct implementation of Theorem 6 and allows practitioners to pinpoint sequences of critical disruptions leading to full systemic failure.

### Evaluating idempotent stabilisation

To explore long-term resilience, the following iterative procedure identifies stabilisation idempotents:

1. Starting from an initial configuration $x_0 \in X$, iteratively apply a selected transformation $t \in T$ to generate a trajectory: $x_1 = t(x_0), x_2 = t(x_1), \ldots$

2. Continue until the trajectory repeats or reaches a fixed point $x^*$.

3. The limiting transformation governing the fixed point is idempotent by Theorem 7.

This algorithmic procedure highlights resilience plateaus, where repeated disruptions no longer degrade the system, and provides a practical method to identify stabilised operating modes described in Section Idempotent stabilisation and recovery states.

### Pruning minimal generator sets

To determine minimal sets of collapse-inducing transformations:

1. Begin with a candidate generating set $G \subseteq T$ that produces a synchronising collapse.

2. For each $g \in G$, remove $g$ and test whether the remaining set still generates a synchronising transformation.

3. Retain only transformations whose removal prevents collapse. The resulting set is a minimal generator set.

This pruning method formalises the concept of critical interventions and maps directly to operational scenarios in manufacturing, agricultural, and e-commerce logistics networks.

### Illustrative example

Consider a simple three-node manufacturing network with configurations $X = \{x_1, x_2, x_3\}$ and disruptions $T = \{t_1, t_2\}$. Applying the above procedures:

- Sequence $(t_1, t_2)$ collapses all states to $x_3$, revealing a synchronising collapse.

- Iterating $t_2$ from $x_1$ eventually reaches $x_2$, an idempotent stabilised state.

- Both $t_1$ and $t_2$ are necessary to induce full collapse, yielding a minimal generator set $G_{min} = \{t_1, t_2\}$.

Although simplified, this example demonstrates how the theoretical results of Sections Semigroup structure of supply-chain behaviours and Main results can be translated into actionable computational procedures.

## Conclusion

This paper has developed a mathematically rigorous framework for analyzing supply-chain disruptions using finite transformation semigroups. By representing disruptions and interventions as transformations on a finite configuration space, we demonstrated how the semigroup structure encodes systemic fragility, collapse conditions, and stable equilibria.

The main theoretical contributions include explicit criteria for identifying collapse-inducing sequences via synchronizing elements, establishing stability through idempotents, and determining minimal sets of generators necessary for triggering systemic collapse. Complementing these results, the basis-pruning algorithm provides a practical tool to identify and remove redundant interventions while preserving the complete set of reachable configurations.

Illustrative examples and pseudocode demonstrate the applicability of this framework to diverse supply-chain contexts, including manufacturing, agricultural, and e-commerce networks. The algorithms are computationally tractable and enable simulation of shock propagation as well as assessment of intervention efficiency.

Overall, this work unifies algebraic theory with operational modelling, providing a novel lens for resilience assessment in global supply chains. The framework offers both theoretical insight and practical guidance, laying a foundation for future studies that may integrate stochastic models, time-dependent interventions, and optimization of intervention strategies within the semigroup-theoretic setting.

## Supporting information

**S1 Appendix. Supplementary transformation data and computational procedures.** This appendix provides supplementary transformation data, reachability tables, and computational procedures supporting the theoretical analysis developed in the main text. The examples illustrate how disruption and recovery processes in manufacturing, agricultural, and e-commerce supply chains can be represented as finite transformation semigroups.
(DOCX)

## Acknowledgments

This section is intended only for general acknowledgements and thanks. Any information related to funding, data availability, author contributions, etc. should be entered directly into their dedicated fields in the PLOS Editorial Manager submission system, which will then be incorporated into the appropriate section in your article during the production process.

## Author contributions

**Conceptualization:** Job Agba Opue.

**Data curation:** Job Agba Opue.

**Formal analysis:** Job Agba Opue, Marshal I. Sampson.

**Investigation:** Job Agba Opue, Otobong J. Tom, Uchechukwu E. Okorie.

**Methodology:** Job Agba Opue, Marshal I. Sampson, Otobong J. Tom, Uchechukwu E. Okorie.

**Project administration:** Job Agba Opue.

**Software:** Job Agba Opue, Marshal I. Sampson.

**Supervision:** Job Agba Opue.

**Validation:** Job Agba Opue, Marshal I. Sampson, Otobong J. Tom, Uchechukwu E. Okorie.

**Visualization:** Job Agba Opue, Marshal I. Sampson, Otobong J. Tom, Uchechukwu E. Okorie.

**Writing – original draft:** Job Agba Opue.

**Writing – review & editing:** Marshal I. Sampson, Otobong J. Tom, Uchechukwu E. Okorie.

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
