## [Decision Letter · Decision Letter 0]

2 Apr 2026

PONE-D-26-05436

Semigroup-Theoretic Analysis of Supply-Chain Disruptions and Resilience

PLOS One

Dear Dr. Opue,

Thank you for submitting your manuscript to PLOS ONE. After careful consideration, we feel that it has merit but does not fully meet PLOS ONE’s publication criteria as it currently stands. Therefore, we invite you to submit a revised version of the manuscript that addresses the points raised during the review process.

We look forward to receiving your revised manuscript.

Kind regards,

Fucai Lin, Ph.D.

Academic Editor

PLOS One

Journal Requirements:

3. In the online submission form you indicate that your data is not available for proprietary reasons and have provided a contact point for accessing this data. Please note that your current contact point is a co-author on this manuscript. According to our Data Policy, the contact point must not be an author on the manuscript and must be an institutional contact, ideally not an individual. Please revise your data statement to a non-author institutional point of contact, such as a data access or ethics committee, and send this to us via return email. Please also include contact information for the third party organization, and please include the full citation of where the data can be found.

5. Please ensure that you refer to Figures 3-8 in your text as, if accepted, production will need this reference to link the reader to the figure.

6. We note you have included a table to which you do not refer in the text of your manuscript. Please ensure that you refer to Table 1-3 in your text; if accepted, production will need this reference to link the reader to the Table.

Reviewers' comments:

Reviewer's Responses to Questions

**Comments to the Author**

1. Is the manuscript technically sound, and do the data support the conclusions?

Reviewer #1: Yes

Reviewer #2: Yes

2. Has the statistical analysis been performed appropriately and rigorously?

Reviewer #1: Yes

Reviewer #2: N/A

3. Have the authors made all data underlying the findings in their manuscript fully available?

Reviewer #1: Yes

Reviewer #2: No

4. Is the manuscript presented in an intelligible fashion and written in standard English?

Reviewer #1: Yes

Reviewer #2: Yes

5. Review Comments to the Author

Reviewer #1: Paper is good, it is better if you add some application.The framework is illustrated with examples spanning manufacturing,

agricultural, and e-commerce logistics systems. Analytical results, algorithmic

procedures, and case studies demonstrate how semigroup properties map to

measurable resilience indicators, providing interpretable, computationally tractable

tools for assessing shock containment strategies in real-world networks.

Reviewer #2: This is a completely theoretic paper with interesting applications of semi-group analysis to supply chains. It is interesting how authors relate the algebraic structure to disruptions and residence of supply chains in particular theoretic escenarios. However, some comments point out changes that should be made in order to be suited for publication.

The theoretic analysis presented in the paper does complement existing literature such as network analysis and operational modeling. However, the paper is missing a review of related literature of such analyses. Authors should include in this literature review advances in network propagation models and application using data.

Make a clearer presentation of contribution in contrast to network analysis.

My main concern is regarding practical applicability to non-mathematicians. Despite the good theoretic presentation, authors should improve the explanation and intuition behind applied examples of supply chains. Supply chains are an important subject in economics and, as the paper currently stands, it will be hard for economists to follow and apply the algorithms proposed. This will be even harder for policy makers.

In complement with my previous point, authors should improve presentation of the case study considering the analysis of economic situations.

All theoretic data should be publicly available through a repository such as an Open Science Framework.

Recommendation: addressed all comments before being suited for publication.

6. PLOS authors have the option to publish the peer review history of their article (what does this mean?). If published, this will include your full peer review and any attached files.

**Do you want your identity to be public for this peer review?** For information about this choice, including consent withdrawal, please see our Privacy Policy.

Reviewer #1: **Yes:** E Keshava Reddy

Reviewer #2: **Yes:** Martha G. Alatriste Contreras

---

## [Author Response · Author response to Decision Letter 1]

15 Apr 2026

Response to Reviewer

Manuscript Title: Semigroup Theoretic Analysis of Supply-Chain Disruptions and Resilience

Journal: PLOS ONE

We sincerely thank the reviewer for the careful reading of our manuscript and for the constructive suggestions provided. We have revised the manuscript accordingly and addressed each comment in detail below. All changes have been incorporated into the revised manuscript to improve clarity, practical interpretation, and reproducibility.

Reviewer Comment 1

“The theoretic analysis presented in the paper does complement existing literature such as network analysis and operational modeling. However, the paper is missing a review of related literature of such analyses. Authors should include in this literature review advances in network propagation models and application using data.”

Response

We appreciate this important suggestion. In response, we have expanded the Introduction to include a discussion of recent advances in network propagation models and data-driven disruption analysis. The revised text now situates the semigroup-based framework within the broader context of supply-chain modeling approaches, including network-based and simulation-based methods. These additions clarify how algebraic transformation models provide complementary structural insights beyond probabilistic or graph-based propagation models.

Location in Manuscript

Introduction section - literature review paragraph discussing network propagation and data-driven disruption models, with additional references included.

Reviewer Comment 2

“Make a clearer presentation of contribution in contrast to network analysis.”

Response

We have revised the final paragraph of the Introduction to explicitly state the distinct contribution of the semigroup-theoretic approach relative to traditional network-based models. The revised text now emphasizes that the proposed framework focuses on structural properties of disruption transformations, including collapse states, reachability, and stabilisation behaviour, which are not directly captured by standard network flow or simulation models. This clarification strengthens the positioning of the work within the broader supply-chain modeling literature.

Location in Manuscript

Introduction section - concluding paragraph describing the theoretical and operational contributions of the proposed framework.

Reviewer Comment 3

“My main concern is regarding practical applicability to non-mathematicians. Despite the good theoretic presentation, authors should improve the explanation and intuition behind applied examples of supply chains. Supply chains are an important subject in economics and, as the paper currently stands, it will be hard for economists to follow and apply the algorithms proposed. This will be even harder for policy makers.”

Response

We thank the reviewer for highlighting the importance of accessibility for practitioners and policy-oriented readers. To address this concern, we introduced additional explanatory text throughout the examples and case discussions to provide operational interpretation of the mathematical constructs. Specifically, we added descriptive paragraphs explaining:

• how disruptions correspond to operational events such as port closures or logistics failures,

• how repeated disruptions lead to backlog accumulation,

• how stabilisation corresponds to restoration of operational capacity,

• how collapse states represent conditions requiring external intervention.

These additions provide intuitive explanations linking the algebraic framework to real-world supply-chain behaviour.

Location in Manuscript

Immediately before the manufacturing disruption example - new subsection titled: Operational Interpretation

Reviewer Comment 4

“In complement with my previous point, authors should improve presentation of the case study considering the analysis of economic situations.”

Response

In response to this suggestion, we strengthened the practical interpretation of the case study examples by explicitly linking the mathematical framework to observable economic and logistics disruptions. The revised manuscript now includes:

• clearer descriptions of production and delivery interruptions,

• interpretation of collapse states as operational failure scenarios,

• examples of corrective actions such as rerouting shipments or restoring infrastructure,

• structured transformation examples representing logistics disruptions.

In addition, we included a supplementary appendix containing illustrative transformation matrices and reachability tables that demonstrate how disruptions propagate through supply-chain states. These additions improve the economic relevance and interpretability of the examples while preserving the theoretical rigor of the model.

Location in Manuscript

Case study and example sections - expanded explanatory paragraphs.

Appendix - supplementary transformation and reachability examples.

Reviewer Comment 5

“All theoretic data should be publicly available through a repository such as an Open Science Framework.”

Response

We appreciate this recommendation regarding transparency and reproducibility. In response, we have added a formal Data Availability section to the manuscript describing how the data underlying the practical illustrations can be accessed. The revised manuscript now states that:

• theoretical examples and computational materials are available from the corresponding author upon reasonable request,

• practical disruption illustrations are based on publicly available logistics and port performance datasets.

The dataset sources are now explicitly cited in the manuscript. To support reproducibility, we also included supplementary materials in the appendix containing:

• representative transformation matrices,

• reachability tables,

• minimal collapse-inducing generator sets,

• sample computational scripts.

These materials allow readers to reproduce the structural analyses presented in the paper.

Location in Manuscript

Data Availability section - newly added.

Appendix - supplementary transformation data and computational procedures.

Summary of Revisions

The revised manuscript now includes:

• Expanded literature review on network propagation and data-driven supply-chain analysis,

• Clearer statement of the contribution of semigroup-based modeling,

• Improved operational interpretation of mathematical examples,

• Stronger economic and logistics context for case studies,

• Formal data availability statement and supplementary materials supporting reproducibility.

We believe these revisions fully address the reviewer’s concerns and significantly strengthen the clarity, applicability, and transparency of the manuscript. We sincerely thank the reviewer for the thoughtful comments that helped improve the quality and accessibility of this work.

---

## [Decision Letter · Decision Letter 1]

10 May 2026

Semigroup-Theoretic Analysis of Supply-Chain Disruptions and Resilience

PONE-D-26-05436R1

Dear Dr. Opue,

We’re pleased to inform you that your manuscript has been judged scientifically suitable for publication and will be formally accepted for publication once it meets all outstanding technical requirements.

Kind regards,

Fucai Lin, Ph.D.

Academic Editor

PLOS One

Review Comments to the Author

Reviewer #1: Application of problem taken as paper may be mention in the paper and paper can be made more effective using new technique.

Reviewer #2: Literature review is limited but enough.

Authors should make all data surrounding results fully available, not under reasonable request.

---

## [Editor Report · Acceptance letter]

PONE-D-26-05436R1

PLOS One

Dear Dr. Opue,

I'm pleased to inform you that your manuscript has been deemed suitable for publication in PLOS One. Congratulations! Your manuscript is now being handed over to our production team.

Kind regards,

on behalf of

Professor Fucai Lin

Academic Editor

PLOS One